# Groundwater Chemical Characteristics and Controlling Factors in a Region of Northern China with Intensive Human Activity

**DOI:** 10.3390/ijerph17239126

**Published:** 2020-12-07

**Authors:** Chaobin Ren, Qianqian Zhang

**Affiliations:** 1School of Chemical Engineering, Zhengzhou University, Zhengzhou 450001, China; robin_ren@gs.zzu.edu.cn; 2School of Civil Engineering, Nanyang Institute of Technology, Nanyang 473004, China; 3Institute of Hydrogeology and Environmental Geology, Chinese Academy of Geological Sciences, Shijiazhuang 050061, China

**Keywords:** groundwater, hydrochemistry, human activity, control factors

## Abstract

The identification of groundwater chemical characteristics and the controlling factors is of major concern in water environment studies. In this study, we identified the groundwater chemical characteristics, evolution laws and main control factors in a region severely affected by human activities using hydrochemical and multivariate statistical techniques. The results showed that the concentrations of NO_3_^−^ and TH were the primary pollution factors in the region with intensive human activity because of high concentration and over the standard rates. The major types of groundwater chemistry were HCO_3_·SO_4_–Ca and HCO_3_·SO_4_–Ca·Mg. The sulfate-type water was as high as 75.0%, 69.2% and 41.2% in the three hydrogeological units. In addition, there were Cl-type and Na-type waters, indicating that the groundwater in this area has been significantly affected by human activities. A principal component analysis (PCA) indicated that the three factors affecting groundwater hydrochemistry in the study area are domestic sewage and fertilizer, water–rock interactions and industrial wastewater. Therefore, we suggest that the government and water environment management departments should prevent the discharge of domestic and industrial wastewater without standardized treatment first in order to effectively prevent the further deterioration of groundwater quality in this area.

## 1. Introduction

Groundwater is one of the most vital resources for human living and production, especially in arid and semi-arid regions [1]. In recent years, with the rapid growth of China’s economy and population, processing capacity of sewage and garbage has not kept pace, and has led to unregulated discharge of sewage (especially in rural areas), the excessive application of pesticide and fertilizer, overexploitation of groundwater, and so on [2,3,4]. This has caused the deterioration of groundwater quality and changed groundwater chemical characteristics. It is well known that changes in the groundwater chemistry not only disrupt many ecological processes [5], but also affect groundwater use. Therefore, fully understanding the evolution law of groundwater chemistry is vital for the sustainable use of groundwater [6,7].

Previous studies found that the evolution of groundwater chemistry is mainly controlled by natural factors (including hydrogeological conditions, aquifer lithology, interactions of groundwater with rock, etc.) and human activities (such as domestic sewage, agricultural fertilizer and overexploitation of groundwater) [8,9]. In recent years, researchers have devoted to study the effect factors on groundwater chemistry [10,11,12]. Huang et al. [8] found that the main factors that control groundwater quality were seawater intrusion, water–rock interactions, lateral flow of river water, sewage discharge and agricultural pollution. Gu et al. [13] identified that the factors influencing groundwater quality were natural mineral dissolution and mine water mixing, domestic sewage and septic tank contamination and agricultural fertilizer contamination.

Selecting the correct analysis method is important for identifying the main factors that control groundwater hydrochemistry. Principal component analysis (PCA) is a mathematical procedure that uses an orthogonal transformation to convert a set of observations of possibly correlated variables into a set of linearly uncorrelated variables called principal components [3]. Therefore, we could identify the unobserved latent controlling factors of groundwater chemistry. Nowadays, PCA has been certified as a powerful tool for determining the controlling factors of groundwater hydrochemistry [14,15] and identifying pollution sources of groundwater quality [16,17]. 

The Hutuo River alluvial-pluvial fan is located in the western portion of the North China Plain. It is an important source of water in Shijiazhuang, and is one of the important receiving areas of the South–North Water Transfer Project. Thus, the groundwater quality directly affects the drinking water security of the residents in the area. However, with rapid urbanization, industrialization and population growth, the groundwater chemistry has been seriously affected by human activities (such as unregulated discharge of domestic sewage and industrial wastewater and overexploitation of groundwater) [18]. Therefore, it is significant for deepening the understanding of groundwater hydrochemical evolution to study the controlling factors of groundwater hydrochemical evolution in this region. The main aims of this study were to: (1) evaluate the characteristics of groundwater hydrochemistry in a region with intensive human activity; (2) analyze the major ion sources using hydrochemistry and multivariate statistical techniques; (3) identify main factors of controlling groundwater chemistry using PCA. The results could be helpful in developing effective water quality protection strategies and could deepen the understanding of groundwater hydrochemical evolution law in a region with intensive human activity.

## 2. Materials and Methods 

### 2.1. Study Area 

The study area is located in the Piedmont inclined plain at the eastern foot of Taihang Mountain, starting from Gangnan Reservoir in the west (the upstream recharge area of Hutuo River alluvial-proluvial fan) to Gaocheng in the east, and extending 5 km from Hutuohe River to the north and south sides. The geographical coordinates are 113°51′05″–114°56′00″ and 37°51′00″–38°24′40″. The administrative area includes Shijiazhuang City, Gaocheng District, Luquan District, Lingshou County, Zhengding County, Wuji County and Pingshan County, with a total area of 2442 km^2^ (Figure 1) and a total population of about 5.5 million. It has a temperate semi-humid and semi-arid monsoon climate, with average temperature and rainfall of 13.3–15.0 °C and 400–750 mm, respectively [1]. The topography is high in the west and low in the east. The landform is an alluvial-pluvial fan group formed by alluvial-pluvial action, and is a sloping plain formed by river and lake facies deposition. The western part is a hilly area with low and medium mountains, and the eastern part is flat North China Plain. The terrain slope is steeper in the west (5–6‰) and other areas are gentler (less than 2‰). The land use types in the study area include farmland (40%), city (11.60%), surface water (2.35%), town (13.09%), and wasteland (mainly including mountains and wasteland) (34.34%) (Figure 1).

The aquifer in the study area belongs to the Quaternary thick aquifer system of Hebei Plain. The stratigraphic lithology is mainly composed of mild clay, sandy loam, sandy mild clay and pebbles, gravel and sand with different particle sizes [19]. The western part of the aquifer is made of a thick layer of sandy gravel; the sandy gravel in the eastern part becomes finer and eventually transitions to mainly coarse sand (Figure 2). The aquifer in the study area is rich in water, and the flow direction of groundwater is the same as that of Hutuo River, which flows from the southwest to northeast. The groundwater mainly consists of bedrock fissure water and loose rock pore water, and the main recharge sources are rainfall vertical infiltration, surface water lateral seepage and flood lateral seepage in rainy season [20]. The study area can be divided into three hydrogeological units, namely the fissure pore water unit in the plain of Hutuo River valley between Gangnan and Huangbizhuang reservoir (HRVP), the hydrogeological unit at the top of Hutuo River alluvial-pluvial fan (THRVPF) and the hydrogeological unit in the central of Hutuo River alluvial-pluvial fan (CHRVPF). The HRVP is located in the transition zone from the mountain area to the plain area. The groundwater depth is shallow (between 2 and 21 m), and it has good conductivity and water richness. In the THRVPF area, the groundwater depth is about 12–40 m and the water conductivity is good. Currently, the main exploitation aquifer is the second water group due to the first aquifer group has been drained. In the CHRVPF area, the groundwater depth is about 40–50 m and the lithology of the aquifer is mainly coarse sand. The third and fourth water groups are the current exploitation aquifer. In recent years, due to the needs of local social and economic development, the groundwater has been intensively exploited, and two large reservoirs have been built in the upstream of the study area, resulting in the recharge source of groundwater being blocked. The depth of groundwater dropped rapidly. At present, one of the largest descending funnels in China has been formed [20].

### 2.2. Groundwater Sampling and Laboratory Analyses

In October 2015, 46 groups of groundwater samples were collected in the Hutuo River alluvial-pluvial fan area (Figure 1). The groundwater was taken from civil wells and agricultural irrigation wells, with a buried depth of 4.0–50.0 m. One 1.5 L and one 500 mL polyethylene plastic bottle were used for the analysis of anions and cations, respectively. The water samples for cation analysis were added to HNO_3_ to adjust the pH < 2. Before sampling, rinse the sampling bottles with purified water and original water sample for three times, respectively, and put the samples into the refrigerator for test.

The following 16 water chemical parameters were analyzed: pH, total dissolved solids (TDSs), potassium (K^+^), sodium (Na^+^), calcium (Ca^2+^), magnesium (Mg^2+^), nitrate (NO_3_^−^), nitrite (NO_2_^−^), ammonia (NH_3_^+^), bicarbonate (HCO_3_^−^), carbonate (CO_3_^2^^−^), chloride (Cl^−^), sulfate (SO_4_^2−^), iron (Fe), manganese (Mn) and total hardness (TH). 

The values of pH were measured in the field using a HQ40D multiparameter instrument (United States of America). Concentrations of the cations and trace elements (Fe and Mn) in the water samples were measured using an inductively coupled plasma atomic emission spectrometer (Agilent 7500ce ICP-MS, Tokyo, Japan), while anion analyses were carried out using spectrophotometry (Perkin-Elmer Lambda 35, Waltham, MA, USA). TDSs were measured using gravimetric methods, HCO_3−_ was measured using acid–base titration and TH was determined using the ethylene diamine tetraacetic acid titration method. All analyses were carried out at the Groundwater Mineral Water and Environmental Monitoring Center at the Institute of Hydrogeology and Environmental Geology, Chinese Academy of Geological Sciences.

### 2.3. Data Analysis

Since the data obtained in this study did not satisfy the normal distribution, we used the non-parametric Kruskall–Wallis and Mann–Whitney U test methods to analyze the significant differences of the concentration of groundwater hydrochemical parameters in different hydrogeological units and land use types. The Spearman correlation analysis was used to identify the correlation between hydrochemical indicators. The principal component analysis was used to identify the main factors of controlling groundwater hydrochemistry. R (3.6.2) (Core Team, Vienna, Austria) and origin (9.0)(OriginLab Corporation, Northampton, MA, USA) were used for data analysis.

### 2.4. Multivariate Data Analysis

#### Contrast Coefficient Variance

Variance is used to describe the dispersion degree of a random variable to its mathematical expectation (mean). Since the absolute value ranges of the chemical components of groundwater may be very different, it is not possible to compare the variances. Therefore, the ratio of each variable (X_i_) of each sample to the mean value of the variable (X_m_) is calculated first, which is defined as the contrast coefficient v, Vi = X_i_/(X_m_) Since the average value of the contrast coefficient is 1, each variable is converted into another variable with the same mean (its contrast coefficient), and then the variance of the contrast coefficient of each variable is calculated to compare the variance.

The calculation formula of contrast coefficient variance (Vσ_i_^2^) is as follows:$$V{{\sigma_{i}}^{2}}^{\;}=\;\frac{1}{n}\sum_{i=1}^{n}{\left(\mathrm{Vi}-\mathrm{Vmi}\right)}^{2}$$
where: Vi is the contrast coefficient of the i-th chemical component of groundwater; Vmi is the average value of the contrast coefficient of the i-th chemical component of groundwater.

## 3. Results 

### 3.1. Main Ionic Characteristics 

The groundwater hydrochemical characteristics of the Hutuo River alluvial-pluvial fan are shown in Table 1. The pH of the groundwater in the study area is 7.38–8.40, and the average values in three hydrogeological units (valley plain area, top of alluvial-pluvial fan and central of alluvial-pluvial fan) are 7.73, 7.80 and 7.63, respectively. There is no excessive water sample, and the groundwater is weak alkaline water. The TDSs value is 152.2–1913 μs/cm. The mean values and over-standard rates of the three hydrogeological units are 950.4 μs/cm and 25%, 720.4 μs/cm and 11.5%, 412.8 μs/cm and 0%, respectively. The main cation in groundwater is Ca^2+^, with a concentration between 20.6 and 345.4 mg/L. The average values of Ca^2+^ in three hydrogeological units are 211.2, 143.8 and 69.8 mg/L, respectively. The order of cation concentration is Ca^2+^ > Na^+^ > Mg^2+^ >K^+^. The concentration of NH_4_^+^ is below the detection limit (0.04 mg/L) in three hydrogeological units in the study area. The main anion of groundwater is HCO_3_^−^. Its concentration range is between 141.0 and 429.3 mg/L. The mean concentration of HCO_3_^−^ in three hydrogeological units is 315.7, 311.7 and 249.3 mg/L, respectively. The order of anion concentration is: HCO_3_^−^ > SO_4_^2−^ > NO_3_^−^ > Cl^−^ > NO_2_^−^ in the valley plain area, and HCO_3_^−^ > SO_4_^2−^ > Cl^−^ > NO_3_^−^ > NO_2_^−^ in the top and middle of alluvial-pluvial fan. CO_3_^2−^ was not detected in the three hydrogeological units in the study area. It is worth noting that the concentration of NO_3_^−^ is 1.76–465.0 mg/L, and the average concentrations in the three hydrogeological units are 176.2, 75.8 and 9.10 mg/L, with over-the-standard (the Grade III standard for groundwater quality in China [21]) rates of 62.5%, 34.6% and 0%, respectively. It is obvious that groundwater in the valley plain area and the top of the alluvial-pluvial fan has been seriously polluted by NO_3_^−^. The concentration range of TH is 76.1–1058 mg/L. The average concentrations of TH in the three hydrogeological units are 650.6, 507.3 and 278.0 mg/L, respectively. The over-the-standard (the Grade III standard for groundwater quality in China [21]) rates are 87.5%, 73.1% and 0%, which is the most important pollution factor in the groundwater in this area. In addition, the concentration range of Fe is 0.01–1.313 mg/L. The average concentrations of Fe in the three hydrogeological units are 0.148, 0.067 and 0.214 mg/L, respectively, and the over-the-standard rates are 12.5%, 3.8% and 8.3%, respectively. 

The spatial variation of hydrochemical indicators is relatively obvious: Hutuo River valley plain area > top of alluvial-pluvial fan > central of alluvial-pluvial fan, which may be related to the groundwater level depth, the lithology of aeration zone and the intensity of human activities [1]. From the valley plain area to the middle of the alluvial-pluvial fan, the buried depth of the groundwater level increases gradually, the particles in the aeration zone become thinner and the permeability becomes poor, which makes it difficult for pollutants to enter the groundwater and the concentrations of the hydrochemical indexes are relatively low.

### 3.2. Hydrochemical Type

It can be seen from Figure 3 that the main hydrochemical types of groundwater in the study area are HCO_3_·SO_4_-Ca and HCO_3_·SO_4_-Ca·Mg, accounting for 21.7% and 28.3%, respectively. The proportions of SO_4_-type water in the three hydrogeological units (valley plain area, top and middle of alluvial-pluvial fan) are 75.0%, 69.2% and 41.2%, respectively. In addition, Cl-type water also accounts for certain proportions in the three hydrogeological units, which are 50.0%, 26.9% and 8.3%, respectively. In this study, the proportions of HCO_3_-Ca(Mg)-type in the three hydrogeological units are 0%, 11.5% and 41.7%, respectively. 

### 3.3. Analysis of Hydrochemistry

A Gibbs diagram is often used to reflect the main control effects of groundwater hydrochemical types (rock weathering, evaporation concentration and precipitation) [22]. The water samples that are greatly affected by atmospheric precipitation are located in the lower right corner of Gibbs figure; the samples that are mainly controlled by rock weathering are located in the middle left corner and the samples; those that are controlled by evaporation crystallization are located in the upper right corner [23]. It can be seen from Figure 4 that most of the water samples from different hydrogeological units fall in the rock weathering area, indicating that rock weathering is the main control factor of groundwater hydrochemistry in this area, and it is less affected by evaporative concentration and atmospheric precipitation.

## 4. Discussion

### 4.1. Groundwater Chemical Characteristics in a Region with Intensive Human Activity 

Groundwater chemical characteristics are influenced by long-term geological history, seepage migration process and human activities [24]. Nowadays, China has experienced rapid urbanization and industrialization, and the influence of human activities on groundwater chemistry is increasing year by year. In the Hutuo River alluvial-pluvial fan region, the groundwater has been intensively exploited and has formed one of the largest descending funnels in China [20]. In addition, the main land use type is farmland in this area—thus, agricultural fertilization and pesticide spraying had serious impact on groundwater chemistry [25]. According to our research, TH and NO_3_^−^ of the groundwater had significantly exceeded the Grade III standard for groundwater quality in China. Thus, TH and NO_3_^−^ of groundwater in the study area have been significantly affected by human activities, such as domestic sewage and agricultural fertilizer [25,26]. This result is consistent with previous research. Zhang et al. (2019) found that NO_3_^−^ was the main impact indicators for poor-quality groundwater in urbanized areas [27]. González Pérez et al. (2020) also found that intensive human activities have serious affected the level of NO_3_^−^ in groundwater [28]. Egbi et al. (2020) also confirmed that the sources of NO_3_^−^ in groundwater primarily originate from human contributions in the Lower Volta River basin of Ghana [29]. Therefore, TH and NO_3_^−^ were the most important parameters, indicating that groundwater is affected by human activities.

SO_4_^2−^ is also a vital parameter that reflected the effect of human activities on groundwater chemistry [30]. Indeed, in this study region, especially in the HRVP area, the mean concentration and over-the-standard (the Grade III standard for groundwater quality in China) rate of groundwater SO_4_^2−^ were 176.7% and 12.5%, respectively. Previous research has confirmed that SO_4_^2−^ was an important indicator in highly anthropogenic regions, which can evaluate groundwater vulnerability and pollution risk [31]. Zhang et al. (2020) also found that the contribution proportion of sewage to SO_4_^2−^ in groundwater was 35.5%–42.7% in the Hutuo River basin [18]. Therefore, SO_4_^2−^ is also a vital indicator representing the effects of human activity.

In addition, the main hydrochemical type of groundwater in this study area was HCO_3_·SO_4_-Ca(Mg). However, previous studies have found that, in 1950, the hydrochemical type of groundwater in the Shijiazhuang area was mainly HCO_3_-Ca(Mg) [24]. In recent years, the hydrochemical type of groundwater has changed to HCO_3_·SO_4_-Ca(Mg), and there is Cl-type water and Na-type water. It can be seen that the groundwater in this area has been significantly affected by human activities, which could influence the safety of the drinking water.

### 4.2. Analysis of Major Ion Sources

#### 4.2.1. Correlation Analysis

If there is a strong correlation between the chemical components in the water environment, it means that they have a common material source. Based on the correlation analysis of groundwater quality parameters in the study area (Table 2), we found that pH had a significant correlation with other indicators. However, TH and TDS showed significant positive correlations with Na^+^, Ca^2+^, Mg^2+^, NO_3_^−^, Cl^−^, HCO_3_^−^ and SO_4_^2−^. It indicated that these components may come from common material sources, such as domestic sewage, agricultural fertilizer, industrial wastewater [25,29]. Because it was in the rainy season, during the process of rainwater recharge, a large number of pollutants from the surface were carried and infiltrated into the aquifer, which promoted water–rock interactions, and increased the concentration of groundwater ions [1]. K^+^ was only positively correlated with Na^+^. Na^+^, Ca^2+^, Mg^2+^, Cl^−^, NO_3_^−^, HCO_3_^−^ and SO_4_^2−^ all showed significant positive correlations, indicating that they came from the same source, such as agriculture runoff, urban runoff and water–rock interactions [8,25]. The correlation between Fe and Mn was very significant, but they did not show correlation with other indexes, which proved that they have a common source.

#### 4.2.2. Source Judgment by Using Contrast Coefficient Variance

The factors that control the chemical composition of groundwater include natural conditions (such as water–rock interactions) and human activities. Under natural conditions, the chemical composition of groundwater will not change much in a complete groundwater system. However, the hydrochemical composition of a groundwater system affected by human activities often shows a great degree of change [32]. Due to the different impact characteristics on the chemical components of groundwater, the contrast coefficient variance is used in this study to distinguish whether the formation of the chemical components of groundwater is mainly dominated by the evolution of natural conditions or by human activities, and to compare the degree of influence of each component by human activities.

On the whole, the variance values of the contrast coefficients of pH, TDS, TH, Ca^2+^, Mg^2+^ and HCO_3_^−^ in the three hydrogeological units are small (Table 3). They are mainly controlled by the evolution of natural hydrochemistry. Because of the shallow groundwater depth, the Hutuo River valley plain unit is greatly affected by human activities and the variance value of the contrast coefficient of each component in groundwater is greater than that of the top and middle hydrogeological units of the Hutuo River alluvial-pluvial fan. It showed that: in the valley plain area, σ^2^ (K^+^), σ^2^(Na^+^), σ^2^(Cl^−^), σ^2^(NO_3_^−^), σ^2^(Fe) and σ^2^(Mn) have large values (σ^2^ > 0.5), which indicated that these components are greatly affected by human activities. In the top of the alluvial-pluvial fan, the values of σ^2^(Na^+^), σ^2^(Cl^−^), σ^2^(Mn) and σ^2^(Fe) are higher (σ^2^ > 0.5); in the middle of the alluvial-pluvial fan, the values of σ^2^(Na^+^), σ^2^(Cl^−^), σ^2^(NO_3_^−^) and σ^2^(Fe) exceeded 0.5, demonstrating that these components are significantly affected by human activities. In the three hydrogeological units, the variance value of the SO_4_^2−^ contrast coefficient is moderately low—0.199, 0.245 and 0.310, respectively—illustrating that it is less affected by human activities and mainly controlled by the evolution of natural hydrochemistry. 

### 4.3. Controlling Factors of Groundwater Chemical

In order to prevent groundwater pollution, the main influencing factors must be identified accurately. In this study, we selected 13 water quality parameters (pH, TDS, TH, K^+^, Na^+^, Ca^2+^, Mg^2+^, NO_3_^−^, Cl^−^, SO_4_^2−^, HCO_3_^−^, Fe and Mn), and identified the main control factors of groundwater pollution in the Hutuo River alluvial-pluvial fan area by PCA. Before principal component analysis, a Kaiser–Meyer–Olkin (KMO) test and Barlett spherical test were carried out on all the data. The results showed that the KMO value was 0.713 and Barlett spherical test value was 963.07 (*p* < 0.001), indicating that the original data were suitable for PCA. Based on the eigenvalue being greater than 1, three main control factors (Figure 5) that caused groundwater pollution in the Hutuo River alluvial-proluvial fan area were identified. These factors can explain 74.01% of all variables, indicating that they basically contain all the information of the above 13 indicators.

The contribution rate of the first main factor (FC1) was TH, Ca^2+^, NO_3_^−^, TDS, Na^+^, K^+^ and Cl^−^, showing a strong positive correlation with FC1, and SO_4_^2−^ showed a moderate positive correlation with PC1. It can be seen that PC1 reflects that groundwater is mainly affected by human activities. The possible sources of Cl^−^ in nature were animal manure, domestic sewage, rising saltwater and road snow melting salt [33]. Since there was no rising saltwater that intruded into the aquifer in the study area, and it was in autumn without snowfall, it can be surmised that neither rising saltwater nor snowmelt salt were the main sources of Cl^−^ in the groundwater in the area. Therefore, the higher concentration of Cl^−^ is most likely to come from animal manure and domestic sewage. The main sources of NO_3_^−^ included domestic sewage, agricultural fertilizers, manure and atmospheric deposition [34]. Previous studies have found that the nitrate pollution of groundwater is mainly related to the use of agricultural fertilizers, discharge of domestic sewage and sewage irrigation [35,36]. During the investigation and sampling, it was found that in the villages at the top of the Hutuo River alluvial-pluvial fan, a large amount of domestic sewage was directly discharged into the Hutuo River channel. In addition, the lithology of the stratum at the top of the fan was relatively coarse, and the sewage was easily infiltrated into the aquifer. As a result, domestic sewage was the main influencing factor of the groundwater in the study area. Furthermore, the main land use type was agricultural land (38.62%), so agricultural fertilizers were also an important source. In conclusion, we can conclude that the first factor (PC1) controlling groundwater chemistry in this area is human activity (domestic sewage and agricultural fertilizer).

The contribution rate of the second main factor (FC2) was 13.78%. HCO_3_^−^ and Mg^2+^ showed a strong positive correlation with PC2, pH showed a medium negative correlation with PC2 and SO_4_^2−^ showed a medium positive correlation with PC2. In general, HCO_3_^−^, Mg^2+^ and SO_4_^2−^ in groundwater were mainly from natural sources (dissolution of dolomite and limestone) [33]. Based on the previous analysis of the contrast coefficient, HCO_3_^−^, Ph, Mg^2+^ and SO_4_^2−^ in the groundwater of the study area were mainly controlled by the chemical evolution of natural water. In recent years, with the development of industry and agriculture, the groundwater in the Hutuo River alluvial-pluvial fan area has been intensively exploited. In addition, two large reservoirs have been built in the upstream of the study area, which cut off the recharge source of groundwater in the area, and continuously reduced the groundwater level. A depression funnel was formed in some parts, breaking the original balance of the groundwater dynamic field, thus triggering a series of physicochemical reactions. The equilibrium is conducive to the dissolution of carbonate minerals (calcite and dolomite), and then the concentration of HCO_3_^−^, Ca^2+^ and Mg^2+^ increased [1]. Therefore, the second factor controlling groundwater in this area is a natural factor (water–rock interaction).

The contribution rate of the third main factor (FC3) is 8.77%. Fe and Mn have strong positive correlations with PC3. In the Hutuo River alluvial-pluvial fan area, the average concentrations of Fe in the groundwater of three hydrogeological units are 0.148, 0.067 and 0.214 mg/L, respectively, and there were four samples that exceeded the national groundwater quality standard of 0.3 mg/L. Previous studies have found that high concentrations of Fe in the water environment were mainly from industrial pollution [37,38]. There was a steel factory upstream of the study area, and the sampling point near it (5) has the highest Fe concentration (0.552 mg/L). The sampling point near Gaocheng industrial park (35) downstream of the study area had an Fe concentration of 1.313 mg/L, which further confirms our hypothesis. Therefore, PC3 shows that the groundwater in the study area is polluted by industrial sewage. Thus, the third factor controlling the groundwater in the study area is the human activity (industrial sewage).

## 5. Managerial Implications and Future Research

The results indicate that high concentrations of NO_3_^−^ and TH in groundwater are present in the region with intensive human activity. Local governments and environmental managers should pay due attention to NO_3_^−^, which can cause not only eutrophication and toxic algal blooms [39], but also increase the risk of diseases, such as methemoglobinemia, blue baby syndrome and stomach cancer [40]. Therefore, groundwater in the region must be treated before drinking. In addition, the main factors controlling the groundwater hydrochemistry in this area are domestic sewage, industrial sewage and agricultural fertilizer. Thus, it is suggested that the government and water environment management department should first stop the substandard discharge of domestic and industrial sewage, and promote optimization of fertilization strategies (such as formula fertilization based on soil testing) in order to effectively prevent the further deterioration of groundwater quality in this area. 

In this study, we have identified the main pollutants and factors controlling groundwater hydrochemistry in the region with intensive human activity. However, we have not quantitatively identified the source of the various pollutants. Nowadays, a quantitative traceability technique combined with isotope and source apportionment models (SIAR model) has been widely used in water environments. For example, previous researchers combined the isotope NO_3_^−^ (δ^15^N and δ^18^O) and SIAR model to identify the NO_3_^−^ sources [41], and used the isotope SO_4_^2−^ (δ^34^S and δ^18^O) and SIAR model to identify the SO_4_^2−^ sources [42]. Therefore, in order to formulate effective controlling measures for controlling groundwater pollution, it is necessary to quantitatively identify the sources of pollutants in groundwater in this region.

## 6. Conclusions

In this research, we applied hydrochemistry (Piper and Gibbs diagrams) and multivariate statistical techniques (variance of contrast coefficient, correlation analysis and PCA) to study the hydrochemistry characteristics and the main control factors of groundwater in the region with intensive human activity—the Hutuo River alluvial-pluvial fan of northern China. The results showed that the groundwater quality was relatively poor, among which the concentration range of NO_3_^−^ and TH are 1.76–465.0 mg/L and 76.1–1058 mg/L, respectively. The over-the-standard rates in three hydrogeological units are 62.5%, 34.6% and 0%, and 87.5%, 73.1% and 0%, respectively, which were the most important parameters indicating that groundwater is affected by human activities. Rock weathering is the main control function of groundwater hydrochemistry in this area. The main types of groundwater hydrochemistry are HCO_3_·SO_4_–Ca and HCO_3_·SO_4_–Ca·Mg, accounting for 21.7% and 28.3%, respectively. SO_4_-type water in three hydrogeological units is as high as 75.0%, 69.2% and 41.2%, respectively. In addition, there are Cl- and Na-type waters, indicating that the groundwater in this area has been significantly affected by human activities. The variance analysis of contrast coefficient shows that K^+^, Na^+^, NO_3_^−^, Cl^−^, Fe and Mn are greatly influenced by human activities, and the other indexes are mainly influenced by natural causes.

The results of PCA show that the three factors controlling groundwater hydrochemistry in the study area are domestic sewage and chemical fertilizer (human activity), water–rock interactions (natural factor) and industrial sewage (human activity). In summary, the main factors controlling the groundwater hydrochemistry in this area are domestic sewage, industrial sewage and agricultural fertilizer. Therefore, in order to effectively prevent the further deterioration of groundwater quality in this area, it is suggested that the government and the water environment management department should first stop the substandard discharge of domestic and industrial sewage and promote optimization of fertilization strategies. In addition, it is necessary to quantitatively identify the sources of pollutants in the future in order to formulate effective controlling measures for controlling groundwater pollution.

## Figures and Tables

**Figure 1 ijerph-17-09126-f001:**
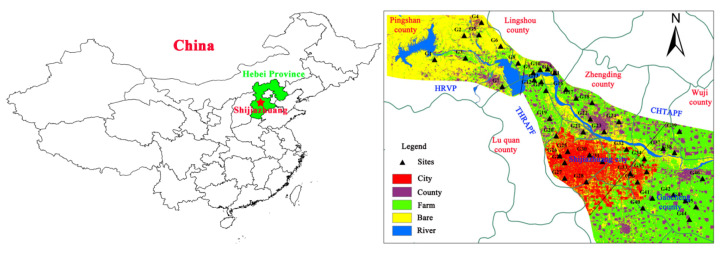
Monitoring sites of groundwater in Hutuo River alluvial-pluvial fan.

**Figure 2 ijerph-17-09126-f002:**
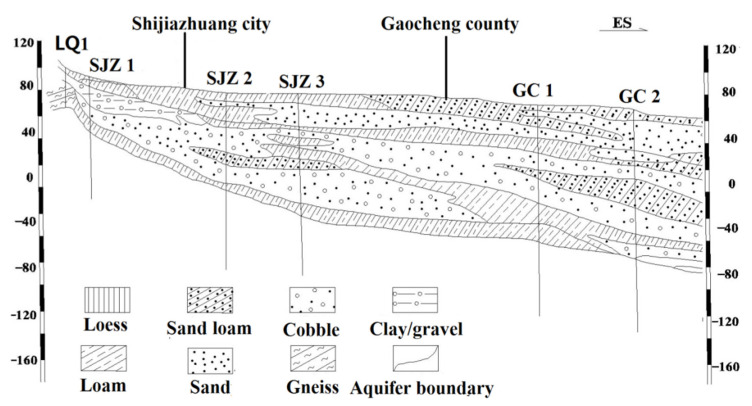
Hydrogeological cross-section of Hutuo River pluvial fan.

**Figure 3 ijerph-17-09126-f003:**
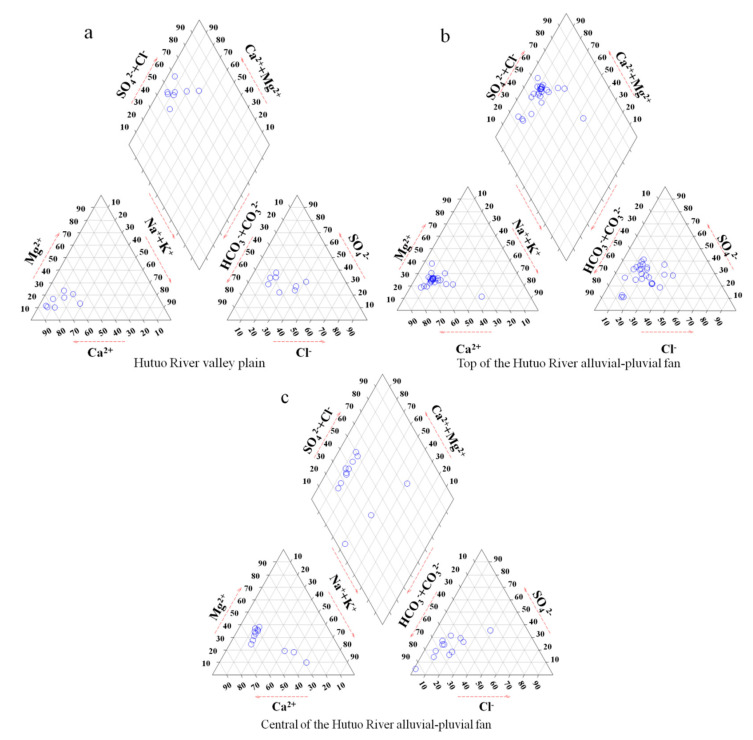
Piper diagram of groundwater in Hutuo River alluvial-pluvial fan.

**Figure 4 ijerph-17-09126-f004:**
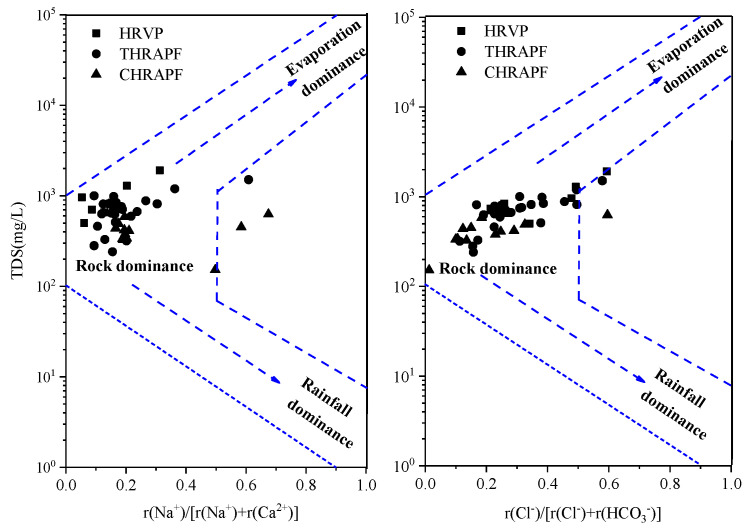
Gibbs diagram for the groundwater in Hutuo River alluvial-pluvial fan.

**Figure 5 ijerph-17-09126-f005:**
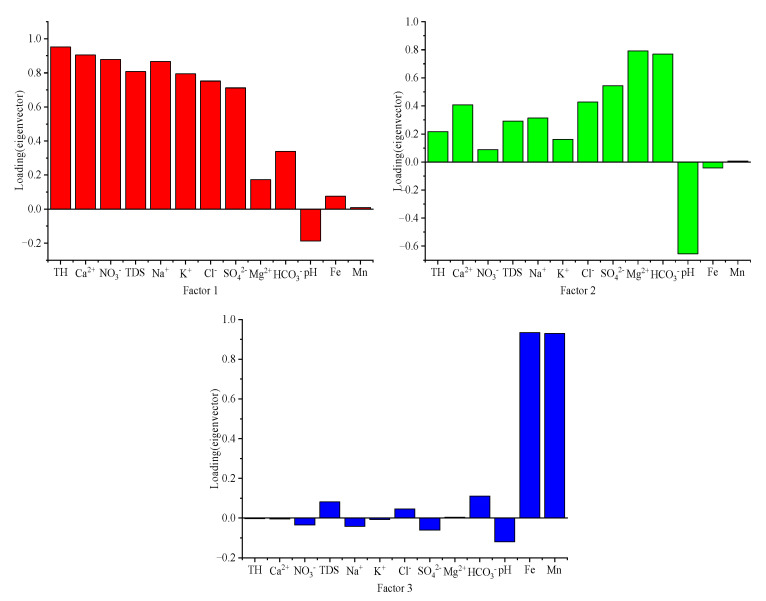
Loadings of 13 selected variables on varimax rotated factors in the Hutuo River alluvial-pluvial fan.

**Table 1 ijerph-17-09126-t001:** Descriptive statistics of groundwater chemical parameters in three hydrogeological units in Hutuo River alluvial-pluvial fan.

Parameters		pH	TDS	K^+^	Na^+^	Ca^2+^	Mg^2+^	HCO_3_^−^	Cl^−^	SO_4_^2−^	NO_3_^−^	NO_2_^−^	Fe	Mn	TH
(I) Hutuo River valley plain unit (n = 8)	Min	7.64	500.7	0.6	10.1	135.2	10.23	187.0	51.09	62.4	68.8	0.002	0.017	0.001	379.8
	Max	7.88	1913	22.0	180.8	345.4	53.1	381.4	322.4	322.3	465.0	0.007	0.552	0.012	1058
	Mean	7.73	950.4	5.80	50.9	211.7	29.7	315.7	121.5	176.7	176.2	0.003	0.148	0.004	650.6
	SD	0.08	455.6	7.66	56.1	68.3	15.8	70.0	99.3	78.7	139.7	0.002	0.180	0.004	219.7
	Exceed standard rate (%)	0	25	—	0	—	—	—	12.5	12.5	62.5	0	12.5	0	87.5
(II) Top of the Hutuo River pluvial fan (n = 26)	Min	7.42	241	0.27	9.08	53.5	11.7	171.7	15.9	21.8	5.04	0.002	0.010	0.001	182.2
	Max	8.40	1508	7.05	307.2	224.1	79.1	429.3	343.5	318.2	216.0	0.041	0.324	0.013	814.2
	Mean	7.80	720.4	2.00	47.3	143.8	36.0	311.7	89.9	151.4	75.8	0.005	0.067	0.003	507.3
	SD	0.22	280.7	1.26	58.0	41.6	14.6	68.8	72.6	74.9	45.0	0.009	0.084	0.004	157.7
	Exceed standard rate (%)	0	11.5	—	3.8	—	—	—	3.8	3.8	34.6	0	3.8	0	73.1
(III) Center of the Hutuo River pluvial fan (n = 12)	Min	7.38	152.2	1.26	17.7	20.6	5.95	141.0	1.06	5.27	1.76	0.002	0.010	0.001	76.1
	Max	7.97	626.5	2.56	139	110.1	41.5	334.2	135.3	168.6	21.8	0.004	1.313	0.040	445.9
	Mean	7.63	412.8	2.13	36.5	69.8	25.2	249.3	42.6	81.4	9.10	0.002	0.214	0.005	278.0
	SD	0.18	124.9	0.50	37.1	22.3	10.8	56.13	35.0	45.3	7.35	0.001	0.399	0.011	96.0
	Exceed standard rate (%)	0	0	—	0	—	—	—	0	0	0	0	8.3	0	0
Standard		6.5–8.5	1000	—	200	—	—	—	250	250	88.6	3.29	0.3	0.1	450

Note: Mean: average value; SD: standard deviation; Min: minimum value; Max: maximum value; Standard is grade III standard for groundwater quality in China (GB/T14848–2017).

**Table 2 ijerph-17-09126-t002:** Spearman’s correlation matrix of water quality of groundwater in the Hutuo River alluvial-pluvial fan.

Parameters	pH	TDS	K^+^	Na^+^	Ca^2+^	Mg^2+^	Cl^−^	SO_4_^2−^	HCO_3_^−^	NO_3_^−^	Fe	Mn	TH
pH	1.000												
TDS	−0.079	1.000											
K^+^	−0.066	0.215	1.000										
Na^+^	−0.088	0.687 **	0.320 *	1.000									
Ca^2+^	−0.067	0.917 **	0.060	0.439 **	1.000								
Mg^2+^	−0.177	0.752 **	0.196	0.540 **	0.648 **	1.000							
Cl^−^	−0.108	0.863 **	0.170	0.706 **	0.741 **	0.598 **	1.000						
SO_4_^2−^	0.015	0.826 **	0.231	0.709 **	0.746 **	0.634 **	0.681 **	1.000					
HCO_3_^−^	−0.165	0.823 **	0.257	0.656 **	0.742 **	0.808 **	0.631 **	0.670 **	1.000				
NO_3_^−^	0.043	0.868 **	0.222	0.370 *	0.909 **	0.585 **	0.683 **	0.649 **	0.645 **	1.000			
Fe	0.148	0.080	0.111	0.051	0.076	0.067	0.063	−0.006	0.127	0.030	1.000		
Mn	0.091	0.032	0.205	−0.018	0.048	−0.012	−0.018	−0.007	0.157	0.029	0.737 **	1.000	
TH	−0.098	0.956 **	0.130	0.519 **	0.964 **	0.803 **	0.770 **	0.756 **	0.820 **	0.889 **	0.083	0.040	1.000

Note: * *p* < 0.05; ** *p* < 0.01.

**Table 3 ijerph-17-09126-t003:** Variance analysis of contrast coefficient of groundwater hydrochemical compositions in the Hutuo River alluvial-pluvial fan.

Parameters	HRVP	THRAPF	CHRAPF
pH	0.000	0.001	0.001
TH	0.114	0.097	0.119
TDS	0.230	0.152	0.092
K^+^	1.744	0.397	0.056
Na^+^	1.216	1.503	1.033
Ca^2+^	0.104	0.084	0.102
Mg^2+^	0.285	0.165	0.183
Cl^−^	0.668	0.653	0.675
SO_4_^2−^	0.199	0.245	0.310
HCO_3_^−^	0.049	0.049	0.051
NO_3_^−^	0.629	0.353	0.652
Fe	1.474	1.562	3.480
Mn	1.011	1.541	0.292

Note: HRVP: Hutuo Rvier Valley Plain; THRAPF: Top of Hutuo River alluvial-pluvial fan; CHRAPF: Center of Hutuo River alluvial-pluvial fan.

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
