# Peer review of "Groundwater Chemical Characteristics and Controlling Factors in a Region of Northern China with Intensive Human Activity"

_ijerph, 2020, doi:10.3390/ijerph17239126_

Round 1

Reviewer 1 Report

Dear Authors         The article is a local study and is based on standard analyzes of groundwater quality. It does not make a significant contribution to the development of science and should rather be published in the national language to have an impact on the decisions of the government and territorial units responsible for water quality. Publication in a worldwide magazine raises my doubts. Detailed comments: Line 29-32. I do not know if it is safe to write this. This is a marketing failure, I do not know if the Chinese Communist Party will be pleased to read the text that indicates that the described region is in the “3rd world country”. Is it safe for you?:) Line 34-41 This introduction is quite obvious, it is not suitable for a scientific journal but for a popular-science one. Line 69-72 Please comment if the lithological profile is the same in each of the 3 examined areas. The geological structure of alluvia can be very different, there may be zones with poor permeability and good permeability. For hydrogeological analyzes, it is necessary to include geological profiles and cross-sections so that the reader has the opportunity to better understand the flow conditions prevailing there. Line 169. This is the last numbered line, there is no numbering below, so I am not able to refer to the following comments in detail.

Conclusions: It is imperative that this section refers to the public health of the people living and managing the area. Unless supplemented, the article may not be appropriate for publication in the J. Environ journal. Res. Public Health.Moreover, the reference literature is very limited and an element of discussion of the results is missing. The problem of groundwater pollution is widely discussed in literature all over the world. For example, modeling the chemical composition of groundwater using the Phreeqc system (https://www.usgs.gov/software/phreeqc-version-3), or the analysis of water susceptibility to pollution using the DRASTIC model: (https: //www.researchgate. net / publication / 305750060_DRASTIC_assessment_of_groundwater_vulnerability_to_pollution_in_the_Vistula_floodplain_in_central_Poland).There are also a number of other methods for assessing groundwater vulnerability to pollution, the above is just an example. The authors did not write the "discussion" part at all, which makes the article a "technical report" and not a "scientific article".In its current form, this makes the article unsuitable for publication.I am not saying that the authors are to conduct such research, but if the article is to meet the criteria of a scientific publication and not popular science, please respond to the above issues in the "discussion" section, and please fill in other problems indicated above.

Author Response

General comments:

The article is a local study and is based on standard analyzes of groundwater quality. It does not make a significant contribution to the development of science and should rather be published in the national language to have an impact on the decisions of the government and territorial units responsible for water quality. Publication in a worldwide magazine raises my doubts.

Response: Thank you for your suggestions. We have rewritten our paper according to your advices, and we have explained the important of the paper in introduction section in line 54-67. Furthermore, we have added the discussion section and explained the most important parameters indicating groundwater affected by human activities in line 215-245. In addition, we added the managerial implications and future research section to further prove the importance of our article in line 350-369.

Detailed comments:

Line 29-32. I do not know if it is safe to write this. This is a marketing failure, I do not know if the Chinese Communist Party will be pleased to read the text that indicates that the described region is in the “3rd world country”. Is it safe for you?:)

Response: Thank you for your suggestions and concern. We have revised some of the wording in line 29-32. In recent years, China's economy has developed rapidly; however, it also brings some environmental pollution problems (such as air pollution and water pollution). In under-developed rural areas, environmental protection facilities were not fully equipped and people’s awareness of environmental protection was relatively poor, which lead to unregulated discharge sewage and over-fertilization. These phenomena also exist objectively.

Line 34-41 This introduction is quite obvious, it is not suitable for a scientific journal but for a popular-science one.

Response: We have rewritten the introduction section and explained the importance of the paper in introduction section in line 54-67.

Line 69-72 Please comment if the lithological profile is the same in each of the 3 examined areas. The geological structure of alluvia can be very different, there may be zones with poor permeability and good permeability. For hydrogeological analyzes, it is necessary to include geological profiles and cross-sections so that the reader has the opportunity to better understand the flow conditions prevailing there.

Response: Thank you for your valuable suggestions. We have added the figure of the hydrogeological cross-section of Hutuo River pluvial fan and added the information of hydrogeological conditions in three hydrogeological units in line 99-110.

Line 169. This is the last numbered line, there is no numbering below, so I am not able to refer to the following comments in detail.

Response: I'm sorry. But, the uploaded manuscript has line number, and there may a problem when converting a word file to a PDF file.

Conclusions: It is imperative that this section refers to the public health of the people living and managing the area. Unless supplemented, the article may not be appropriate for publication in the J. Environ journal. Res. Public Health. Moreover, the reference literature is very limited and an element of discussion of the results is missing. The problem of groundwater pollution is widely discussed in literature all over the world. For example, modeling the chemical composition of groundwater using the Phreeqc system (https://www.usgs.gov/software/phreeqc-version-3), or the analysis of water susceptibility to pollution using the DRASTIC model: (https: //www.researchgate. net / publication / 305750060_DRASTIC_assessment_of_groundwater_vulnerability_to_pollution_in_the_Vistula_floodplain_in_central_Poland).There are also a number of other methods for assessing groundwater vulnerability to pollution, the above is just an example. The authors did not write the "discussion" part at all, which makes the article a "technical report" and not a "scientific article". In its current form, this makes the article unsuitable for publication. I am not saying that the authors are to conduct such research, but if the article is to meet the criteria of a scientific publication and not popular science, please respond to the above issues in the "discussion" section, and please fill in other problems indicated above.

Response: We have added the discussion section and explained the most important parameters indicating groundwater affected by human activities in line 215-245. In addition, we added the managerial implications and future research section in line 350-369.

Reviewer 2 Report

I made some editing (in red ink), and scanned the edited paper. I have attached a copy to this report.  

Author Response

Reviewer 2

I made some editing (in red ink), and scanned the edited paper. I have attached a copy to this report. 

Response: Thank you for your serious comment, we have revised the grammatical issues in accordance with your request in the uploaded revised Manuscript.

Reviewer 3 Report

Title: Groundwater chemical characteristics and controlling factors in an intensely human activity region of northern China

Dear authors,

The topic of your paper is interesting, however it is a highly studied topic; and for this reason, to contribute something new requires a broad search for information and an approach in which innovative results are obtained. The authors should work to improve this proposal: provide some new data or deepen the discussion on the data already presented (compare them with others in other regions where there is high human activity: China, some regions of the USA, the Mediterranean region, Asia, etc.), discuss differences, similarities and causes of them, and design a suitable article.

Aesthetics and layout should also be reviewed.

There are also certain specific aspects which should be improved. Following are some recommendations for authors to consider:

1.-Introduction:

It is a bit scarce, you must extend the introduction to justify the object of your research. Please argue more why this study may contribute with new knowledge. Regarding originality, some points could be developed in the introduction section: Please, clarify why your paper is important. What are you going to discover? Why is this topic important? Who is going to earn more with your results about groundwater quality?

2.- M&M:

Lines 51-83: References are missing to justify the information provided.

Line 68, 160, 179: There are different fonts and font sizes (Review the layout of the text).

3.- Results and Discussion:

You must develop the research further, add some new variable and contrast the results. You need to provide more data, and work with them ... compare them with the existing literature and conclude what is the new knowledge that your study provides.

A greater contribution of data is necessary, or a greater discussion and comparison with the existing literature.

You should add some new source to compare your: i.e. Gonzalez-Perez et al., 2020 (https://doi.org/10.3390/w12072006): This document determined that different aquifers sited in Ibiza (an island with extremely high human activity in Spain) present excessive concentrations of certain ions, NO3 (anthropogenic pollution) and Cl (due to overexploitation).

Or i.e. Egbi et al, 2020 (https://doi.org/10.1016/j.ecoenv.2020.110227) sited in Volta River Basin (Ghana).

4.- Conclusions

Limited findings and discussion section relating to the literature. Even the study results were not fully discussed.

This paper about groundwater vulnerability and risk assessment presents interesting information, maybe it will help you: Busico et al., 2017 (https://doi.org/10.1016/j.scitotenv.2017.07.257).

It would be interesting some additional managerial implications in line with the findings of the study. Practical implications?  Something to inspire future research or implications for practice. Public policies that should be derived from your results?

Line 328-330: Irrelevant content.

Best regards,

Author Response

Reviewer 3

The topic of your paper is interesting, however it is a highly studied topic; and for this reason, to contribute something new requires a broad search for information and an approach in which innovative results are obtained. The authors should work to improve this proposal: provide some new data or deepen the discussion on the data already presented (compare them with others in other regions where there is high human activity: China, some regions of the USA, the Mediterranean region, Asia, etc.), discuss differences, similarities and causes of them, and design a suitable article.

Response: Thank you for your valuable suggestions. We have added the discussion section and explained the most important parameters indicating groundwater affected by human activities in line 215-245. In addition, we added the managerial implications and future research section in line 350-369.

Aesthetics and layout should also be reviewed.

Response: We have adjusted the layout of paper.

There are also certain specific aspects which should be improved. Following are some recommendations for authors to consider:

1.-Introduction:

It is a bit scarce, you must extend the introduction to justify the object of your research. Please argue more why this study may contribute with new knowledge. Regarding originality, some points could be developed in the introduction section: Please, clarify why your paper is important. What are you going to discover? Why is this topic important? Who is going to earn more with your results about groundwater quality?

Response: Thank you for your valuable suggestions. We have extended the introduction and explained the importance of the paper in the introduction section in line 47-67. 

2.- M&M:

Lines 51-83: References are missing to justify the information provided.

Response: We have added the references to justify the information provided in line 78, 89 and 95.

Line 68, 160, 179: There are different fonts and font sizes (Review the layout of the text).

Response:  We have revised the font sizes.

3.- Results and Discussion:

You must develop the research further, add some new variable and contrast the results. You need to provide more data, and work with them ... compare them with the existing literature and conclude what is the new knowledge that your study provides.

A greater contribution of data is necessary, or a greater discussion and comparison with the existing literature.

Response:  Thank you for your valuable suggestions. We have added the discussion section and explained the most important parameters indicating groundwater affected by human activities in line 215-245.

You should add some new source to compare your: i.e. Gonzalez-Perez et al., 2020 (https://doi.org/10.3390/w12072006): This document determined that different aquifers sited in Ibiza (an island with extremely high human activity in Spain) present excessive concentrations of certain ions, NO3 (anthropogenic pollution) and Cl (due to overexploitation).

Or i.e. Egbi et al, 2020 (https://doi.org/10.1016/j.ecoenv.2020.110227) sited in Volta River Basin (Ghana).

Response: We have added the literature you recommended and discussed.

4.- Conclusions

Limited findings and discussion section relating to the literature. Even the study results were not fully discussed. This paper about groundwater vulnerability and risk assessment presents interesting information, maybe it will help you: Busico et al., 2017 (https://doi.org/10.1016/j.scitotenv.2017.07.257).

Response: We have added the literature you recommended and discussed. In addition, we have discussed the most important parameters indicating groundwater affected by human activities in line 215-245.

It would be interesting some additional managerial implications in line with the findings of the study. Practical implications?  Something to inspire future research or implications for practice. Public policies that should be derived from your results?

Response: We added the managerial implications and future research section in line 350-369.

Line 328-330: Irrelevant content.

Response: Thank you for your reminder. Our organization requires that the project number must be indicated when the article is published, otherwise, the publication expenses of the article cannot to paid by organization. We have previously published articles in this journal, and project funding was also indicated in this section. Therefore, we put project funding in the section.

Round 2

Reviewer 3 Report

Title: Groundwater chemical characteristics and controlling factors in an intensely human activity region of northern China

Dear authors,

I thank the authors for this review work, which satisfies most of my concerns about the article.

1.-Introduction:

Now OK

2.- M&M:

Now OK

3.- Results and Discussion:

Now OK.

4.- Conclusions

Now OK

5.- References:

Please, pay attention to the numbering of the references. You should assign the number in order of appearance in the text; you must reorder them.

The numbering of the lines disappears after line 189.

Best regards

Author Response

Comments and Suggestions for Authors

Title: Groundwater chemical characteristics and controlling factors in an intensely human activity region of northern China

Dear authors,

I thank the authors for this review work, which satisfies most of my concerns about the article.

1.-Introduction:

Now OK

2.- M&M:

Now OK

3.- Results and Discussion:

Now OK.

4.- Conclusions

Now OK

5.- References:

Please, pay attention to the numbering of the references. You should assign the number in order of appearance in the text; you must reorder them.

Response: Thank you for your reminder. The number of the references was assigned in order of appearance in the text and we have adjusted a few the numbering of the references. However, there were several of the references contain multiple citations, so the numbers of the references seem to be confusing,

The numbering of the lines disappears after line 189.

Response: We have added the numbering of the lines.